# Time-Dependent Behavior of Callovo-Oxfordian Claystone for Nuclear Waste Disposal: Uncertainty Quantification from In-Situ Convergence Measurements

**Duc-Phi Do** [1] , **Ngoc-Tuyen Tran** [2] , **Dashnor Hoxha** [1,*] , **Minh-Ngoc Vu** [3] and **Gilles Armand** [3]

1    INSA CVL, Lamé, EA 7494, University Orléans, University Tours, 45100 Orléans, France; duc-phi.do@univ-orleans.fr
2    Faculty of Engineering-Technology, Hatinh University, Hatinh 480000, Vietnam; tuyen.tranngoc@htu.edu.vn
3    Andra, R&D Division, 92298 Chatenay-Malabry, France; minh-ngoc.vu@andra.fr (M.-N.V.); gilles.armand@andra.fr (G.A.)
*    Correspondence: dashnor.hoxha@univ-orleans.fr; Tel.: +33-2-38-49-43-75

**Abstract:** The sustainability of geotechnical infrastructures is closely linked with their long-time behavior. In fact, there is not a straightforward procedure to predict this behavior, and very often, the back analyses of observed data are the best tool to understand their long-time response. In-situ observations of drifts constructed in the Callovo-Oxfordian (COx) claystone, the potential host formation for geological radioactive waste disposal, in France exhibit a progressive convergence. These convergence measurements with quite significant dispersions reveal a considerable uncertainty of time-dependent behavior of this argillaceous rock that can strongly affect the transmit loading to liners, hence the long term stability of the drift. Consequently, the uncertain quantification of the creep behavior of COx claystone presents an important task before analyzing the safety of the waste disposal system. In this work, this challenge was conducted by using the well-known Bayesian inference technique. For this aim, on the one hand, the effectiveness of the classical and hierarchical Bayesian techniques to quantify the epistemic and aleatoric uncertainties of the time-dependent behavior of the host rock were investigated using synthetic data. On the other hand, we dealt with the uncertain quantification of the Lemaitre parameters that characterize the visco-plastic behavior of COx claystone thanks to the real data of in-situ convergence measurements of drifts.

**Keywords:** uncertainty quantification; Bayesian inference; creep behavior; COx claystone; drift convergence; anisotropy

## 1. Introduction

The world objectives for green energy, and the tensions about organic fuels, are accompanied with efforts for carbon-free alternative energies. In all mitigation paths for sustainable energy development in the future, the Intergovernmental Panel on Climate Change (IPCC) reports on the global warming impact scheduled, among other tools, a significant increase of the world nuclear energy production [1]. While the practice of low and intermediate radioactive waste management has been largely conducted, the long-term behavior of nuclear waste disposal is yet a challenging task and a dynamic field of study. Nowadays, all different solutions in various countries worldwide make use in one way or another of so called "geological barriers" as an element of the proposed solution.

Considered as a potential host formation for geological radioactive waste disposal in France, the Callovo-Oxfordian (COx) claystone has been intensively studied in many research programs led by the French National Radioactive Waste Management Agency (Andra). Among them, the in-situ observations and experimentation in the Underground Research Laboratory in Meuse and Haute Marne (URL M/HM) have been conducted since 2000 to characterize both the short- and long-term behavior of this host formation.

The convergence measurement of drifts within the URL exhibits not only a complex time-dependent behavior but also an anisotropy of the convergence in all drifts [2–4]. Completed by many other studies conducted in laboratories, it has also been shown that the short-term behavior of this host rock seems to be predominant by the elastoplastic and damage mechanisms, while at the long term, the creep behavior is dominant [5]. Further, a high dispersion of the mechanical properties of COx claystone due to their dependence on mineralogical composition has been observed.

Specifically, the measurement in URL highlights that the initial stress state is characterized by three principal stresses: the minor horizontal stress is quasi-similar to the vertical stress ($\sigma_v \approx \sigma_h \approx 12.5$ MPa), while the major horizontal stress is about 1.3 times higher than the minor stress ($\sigma_H \approx 1.3\sigma_h$) [1]. The excavation of drift in both directions of the major and minor horizontal stresses induces a fractured zone. Indeed, for drifts parallel to $\sigma_H$, it was observed that the fractured zone presents a dissymmetrical shape despite the quasi-isotropic stress state in the drift section [2,3]. The extension of fracture is more developed in the horizontal direction of the drift cross section. The observed anisotropy is also confirmed in the convergence measurements that are conducted in the horizontal and vertical directions. The higher convergence, about two times higher, is noted in the horizontal direction, but the evolution in time of both the horizontal and vertical convergence seems similar [2].

To reproduce the observed phenomenon of anisotropic convergence and the dissymmetrical shape of the fractured zone around drift under the quasi-isotropic stress state, different models have been developed to describe the complex short- and long-term behavior of COx claystone. Sophisticated models considering the anisotropy of plasticity, visco-plasticity, or damage have been proposed. However, a large number of parameters are needed in these models, and they are not easy to be determined [6–9]. To overcome this drawback, some scholars [10,11] proposed a simplified approach to simulate the anisotropic elasto-plastic behavior of the host rock. In their model, the anisotropy of fractured zones was considered in the geometrical model of drift by defining explicitly their size and shape, as characterized from in-situ monitoring [3]. These zones have similar elastic properties but lower plastic properties compared to the intact rock. Further, the Mohr Coulomb was chosen to describe the elasto-perfectly plastic behavior of both the fractured zone and the intact rock.

Recently, in a contribution of the present authors [12,13], the stochastic analysis was conducted to quantify the uncertainty effect of the time-dependent behavior of COx claystone on the long-term stability of drift. Following that, the creep behavior of host rock was characterized by the Lemaitre viscoplastic model, while the Kriging metamodeling technique was chosen for the reliability analysis. The obtained results showed that the stability of drift support can strongly depend on the uncertainty of creep behavior of the host rock. However, in this last study, the uncertainty of the viscoplastic behavior of COx claystone was only quantified from the triaxial creep tests performed in the laboratory.

In this work, the effectiveness and the applicability of the well-known Bayesian inference (BI) to quantify the uncertainty of the time-dependent behavior of COx claystone were demonstrated by using the in-situ data provided from the measurements of drift convergence. In comparison with the previous study in [12], the difficulty and complexity of the problem treated in this work relates to the consideration of the excavation damaged zone around drift and its effect on the anisotropic convergence. For this purpose, we shared the same idea as in [10,11] by considering the short-term anisotropic response of drift convergence, in which we imposed ad hoc an elliptical shape of the fractured zone, as characterized in situ. This elliptical zone, having also different plastic properties of the intact rock, induced a redistribution of stress state and hence affected the long-term response of the drift.

In what follows, the BI will be briefly presented in the next section. Then, the efficiency of BI on the uncertainty quantification of creep rock behavior will be investigated using

the synthetic data of tunnel convergence provided from the analytical solution. Finally, the application on the time-dependent behavior of COx claystone behavior is conducted.

## 2. Uncertainty Quantification by Bayesian Inference

### 2.1. Classical Bayesian Inference

The deterministic calibration process of the model's parameters from the experimental data (i.e., inverse problem) has been largely conducted in the literature [14,15]. However, this well-known method cannot capture the uncertainty of the obtained results when only a single value of each parameter (i.e., the best-fit input parameters) can be determined. To overcome this drawback, the stochastic inversion, such as the Bayesian inference, has been intensively undertaken in the last two decades. Considering input parameters as random variables, this statistical inference technique allows for quantifying the associated uncertainty of these parameters, which is crucial for the reliability analysis and for optimization of the design of the structure [12,13,16,17].

Supposing that the vector $y = \{y_1, y_2 \ldots y_N\}$ is a data set of the observations, the random input parameters of the considered model $u(\theta)$ are gathered in the vector $\theta = \{\theta_1, \theta_2, \ldots, \theta_M\}$. The principal idea of BI consists of computing the probabilistic distribution $p(\theta \mid y)$ of the random vector $\theta$ conditional on training data $y$ using the Bayes' theorem [17,18]:

$$p(\theta|y) = \frac{p(y|\theta)p(\theta)}{p(y)} = \frac{p(y|\theta)p(\theta)}{\int p(y|\theta)p(\theta)d\theta} \tag{1}$$

In Equation (1), $p(\theta)$ is the prior distribution of the random variables, which represents the beliefs about $\theta$ *a priori* (i.e., before any data has been observed). Thus, Equation (1) updates the belief about $\theta$ to the posterior by taking into account the observed data in the computation of $p(\theta \mid y)$ to reduce the discrepancy between observation and simulation. The function $p(y \mid \theta)$ and $p(y)$ are referred to as the likelihood and marginal likelihood (or evidence). Assuming that the discrepancy between the observation and the model prediction $u(\theta)$ have a Gaussian distribution with a zero mean value and an unknown variance $\sigma^2$, the likelihood function $p(y \mid \theta)$ is written as:

$$p(y|\theta) = \prod_{i=1}^{N} \frac{1}{\sqrt{(2\pi\sigma^2)^N}} exp\left(-\frac{1}{2\sigma^2}(y_i - u(\theta))^T(y_i - u(\theta))\right) \tag{2}$$

Usually, the posteriori function $p(\theta \mid y)$ described in Equation (1) is implicit; hence, the exploration of this function is difficult. The Monte Carlo (MC) sampling technique is considered the most appropriated method in this case to explore this implicit posteriori function. However, to reduce the required large number of samples in the classical MC method to achieve statistical convergence, the Markov Chain Monte Carlo (MCMC), which is a class of sequential sampling strategies, is commonly used. Following the MCMC method, a Markov chain is constructed when the next sampled state depends on the current state by using the Metropolis-Hastings or Gibbs sampling technique [18–20]. Depending on the shape of the posteriori distribution, the required number of samples in MCMC can remain high. Therefore, in case of a complex structure, the likelihood evaluation in the MCMC-based model uncertainty quantification can be conducted by combining it with the surrogate model that approximates the structure response to reduce the computational cost of the numerical simulation. Some well-known surrogates, based on: Kriging, polynomial chaos expansions (PCEs), and Artificial Neuron Network (ANN), are among the most widely used [6,7,10–14].

Note that, in the uncertainty quantification by the BI process, the construction of the full posteriori probability is not crucial. Instead, one may be interested only on the determination of its low-order moments, such as the mean and the variance. These last

parameters can be estimated by finding the optimal values of $\theta$ that minimize the negative log posterior (known also as the maximum a posteriori *MAP*):

$$\theta_{MAP} = \underset{\theta}{argmin}[-log(p(\theta|y))] \tag{3}$$

*2.2. Hierarchical Bayesian Inference*

The BI, as presented previously, allows for quantifying the uncertainty of the model's parameters by considering only the epistemic uncertainty in a data set. Following that, this probabilistic inversion accounts for the uncertainty related to the lack of knowledge and to measurement errors, while the heterogeneous characteristic of the material model is ignored.

Recently, different scholars developed the hierarchical BI to take into account the inherent variability of the input parameters of the material model [21–23]. In addition to the epistemic uncertainty, the aleatory uncertainty that relates to the spatial variations of the material properties is considered in this approach. Regarding the experimental data, they are presented in multiple sets $y_j$ ($j = 1, 2 \ldots, N_s$), and each data set $y_j = \{y_1, y_2, \ldots, y_{Nj}\}$ of the $j$th experiment describes a realization $\theta_j$ of the random vector $\theta$ of input parameters. Thus, the hierarchical BI consists of quantifying the corresponding unknown statistical moments (i.e., the mean and standard deviation) $\chi = \{\mu_\theta, \sigma_\theta\}$ of the input parameters $\theta$ by considering these so-called hyperparameters as uncertain random variables.

The extension of the classical BI in this context can be conducted by substituting in Equation (2) a corresponding joint posterior function:

$$p(\theta, \chi|y) = \frac{p(y|\theta, \chi)p(\theta, \chi)}{p(y)} = \frac{p(y|\theta, \chi)p(\theta|\chi)p(\chi)}{p(y)} \tag{4}$$

In Equation (4), the joint prior distribution $p(\theta, \chi)$ is described by the input parameters' prior that is conditional on the hyperparameters $p(\theta \mid \chi)$ with their own prior distribution $p(\chi)$.

The hierarchical BI focuses then on the uncertainty quantification of the hyperparameters $\chi$ by integrating out the input parameters $\theta$ (which are considered nuisance parameters) in the marginalized likelihood function [21–23]:

$$p(\theta, \chi|y) = \frac{p(y|\theta, \chi)p(\theta, \chi)}{p(y)} = \frac{p(y|\theta, \chi)p(\theta|\chi)p(\chi)}{p(y)} \tag{5}$$

where:

$$p(y|\chi) = \prod_{j=1}^{N_s} p(y_j|\chi) \tag{6}$$

$$p(y_j|\chi) = \int p(y_j|\theta_j)p(\theta_j|\chi)d\theta_j \tag{7}$$

$$p(y) = \int p(y|\chi)p(\chi)d\chi \tag{8}$$

The calculation of the evidence $p(y_j \mid \chi)$ for each data set $y_j$ conditional on the hyperparameters $\chi$ in Equation (7) presents an important step in the hierarchical BI and different methods have been developed in the literature. For example, Sedehi et al. [21] used the Laplace asymptotic approximation in the integral (Equation (7)), while Nagel and Sudret [22] proposed some advanced MCMC techniques. In their work, Wu et al. [23] approximated the integral by the important sampling method using the proposal distribution $p(\theta_j \mid y_j)$ to reduce the computational cost:

$$p(y_j|\chi) \approx \frac{1}{M_j} \sum_{k=1}^{M_j} \frac{p\left(y_j|\theta_j^{(k)}\right)p\left(\theta_j^{(k)}|\chi\right)}{p\left(\theta_j^{(k)}|y_j\right)} = \frac{p(y_j)}{M_j} \sum_{k=1}^{M_j} \frac{p\left(\theta_j^{(k)}|\chi\right)}{p\left(\theta_j^{(k)}\right)} \tag{9}$$

In Equation (9), the samples $\left\{\theta_j^{(k)} | k = 1, \ldots, M_j\right\}$ are drawn from the posterior distribution $p(\theta_j | y_j)$ of each data set. Thus, as the main advantage, this hierarchical BI comes as a postprocessing of the results of the classical BI conducted for each data set, through which the likelihood $p(y | \chi)$ can be estimated as:

$$p(y|\chi) \approx \prod_{j=1}^{N_s} \left( \frac{p(y_j)}{M_j} \sum_{k=1}^{M_j} \frac{p\left(\theta_j^{(k)} | \chi\right)}{p\left(\theta_j^{(k)}\right)} \right) \tag{10}$$

In this work, this hierarchical BI was chosen and implemented in the well-known Matlab toolbox UQLab [24]. In this process, the classical BI that is based on the sequential sampling MCMC, using the adaptive Metropolis algorithm, was chosen to solve the inverse problem of each data set. In addition, to reduce the computational cost of the numerical simulation, the well-known Kriging metamodeling technique (see [12,13,16,17] for details) was also used to approximate the structure response. More precisely, the Kriging meta-model was constructed from the results of the numerical simulation of 240 samples of the Design of Experiment (DoE) generated by the Latin Hypercube Sampling technique to approximate the tunnel convergence. The postprocessing step was then conducted by moving the posterior samples of each classical BI problem up the likelihood function (Equation (10)) of the hierarchical BI.

## 3. Numerical Applications Using Synthetic Data

BI has been largely used to identify parameters and their associated uncertainty of the time-independent behavior, such as the elastic or elastoplastic behavior of materials [19,25,26]. Its application to quantify the uncertainty of the creep behavior is not much discussed in the literature. For instance, in [27], the authors investigated the capability of BI to identify the uncertainty of the viscoelastic parameters, in which the synthetic data of different experiments (e.g., relaxation and creep experiments, constant strain rate test) are artificially created. The efficiency of the classical BI was discussed, and the higher accuracy of identified values could be improved by increasing the number of measurements.

In this work, the applicability of BI was investigated to quantify the uncertainty of creep rock behavior using the results of in-situ tunnel convergence measurements. For this aim, we firstly used the synthetic data of tunnel convergence, which were artificially created by using the analytical solution of a deep tunnel drilled in an elasto-viscoplastic rock under a hydrostatic stress $P_0$ (Figure 1a). This solution was recently presented in [28], in which the time-dependent behavior of creep rock was characterized by the fractional derivative viscoplastic (FDVP) model. This constitutive model was established from the connection in a series of different components: the fractional-order Maxwell element, the Klevin element, and the Mohr-Coulomb plastic slider. For the sake of simplicity, in this work, we used the simpler FDVP, in which only the fractional-order Maxwell was connected in a series with the Mohr-Coulomb plastic slider (see Figure 1a). In comparison with the contribution in [28], the number of parameters used to characterize the elasto-viscoplastic behavior of rock was lower, being represented by the spring $G_M$, the fractional-order derivation dashpot $\eta_M$ and fractional-order coefficient $\beta$ of Maxwell element, as well as the three well-known parameters of the Mohr-Coulomb model (i.e., the cohesion $C$, the friction angle $\varphi$, and the dilation angle $\psi$).

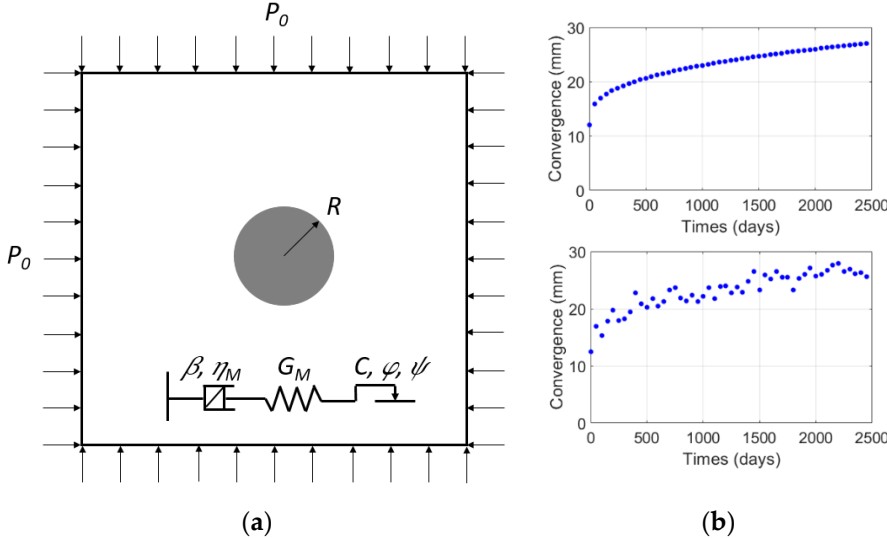

**Figure 1.** (**a**) Circular tunnel in the FDVP rock; (**b**) calculated convergence on the surface of tunnel without and with additive noise.

While the derivation of this analytical solution was detailed in [28], we capitulated as follows the explicit expression of convergence $u_{evp}$ on the surface tunnel:

$$\frac{u_{evp}}{R} = \frac{F_1}{2\left(N_\varphi + N_\psi\right)}\left(\frac{1}{G_M} + \frac{t^\beta}{\eta_M \cdot \Gamma(1+\beta)}\right) + F_2\left(\frac{R_p}{R}\right)^{N_\psi + 1} \tag{11}$$

where:

$$F_1 = \left(1 - N_\psi\right)\left[\left(1 - N_\varphi\right)\left(1 - \lambda\right)P_0 - 2C\sqrt{N_\varphi}\right] \tag{12}$$

$$F_2 = \frac{1}{2}\left[\lambda_e P_0 - \frac{F_1}{N_\varphi + N_\psi}\left(\frac{R_p}{R}\right)^{N_\varphi - 1}\right]\left(\frac{1}{G_M} + \frac{t^\beta}{\eta_M \cdot \Gamma(1+\beta)}\right) \tag{13}$$

$$R_p = R\left[\frac{2}{N_\varphi + 1}\frac{\left(N_\varphi - 1\right)P_0 + 2C\sqrt{N_\varphi}}{\left(1 - \lambda\right)\left(N_\varphi - 1\right)P_0 + 2C\sqrt{N_\varphi}}\right]^{N_\varphi - 1} \tag{14}$$

$$\lambda_e = \frac{1}{N_\varphi + 1}\left(N_\varphi - 1 + \frac{2C\sqrt{N_\varphi}}{P_0}\right) \tag{15}$$

$$N_\varphi = \frac{1+sin(\varphi)}{1-sin(\varphi)}, \quad N_\psi = \frac{1+sin(\psi)}{1-sin(\psi)} \tag{16}$$

In Equations (11) to (16), the parameters $R$ and $\lambda$ are the radius of the circular tunnel and the deconfinement ratio, while $\Gamma(\beta)$ is the well-known gamma function, with respect to the fractional-order coefficient $\beta$.

The chosen values of the input parameters to calculate the tunnel convergence using Equation (11) are summarized in Table 1. The additive noise characterized by a normal distribution with a zero mean and a standard deviation of $\sigma_n = 1$ (mm) was generated in the convergence of the tunnel to create the synthetic data for the uncertainty quantification (Figure 1b). Totally, a data set with 50 values of tunnel convergence were artificially generated in the range of 2500 days.

**Table 1.** Chosen parameters for the calculation of synthetic data.

| $G_M$ (GPa) | $\eta_M$ (GPa.year) | $C$ (MPa) | $\varphi$ (°) | $\psi$ (°) | $\beta$ | $P_0$ (MPa) | $R$ (m) | $\lambda$ |
|---|---|---|---|---|---|---|---|---|
| 1.73 | 3.06 | 6 | 20 | 0 | 0.35 | 12.5 | 2.6 | 1 |

Using the synthetic data, the classical BI was conducted to identify the time-dependent behavior parameters characterized by the two parameters $\eta_M$ and $\beta$, while the other mechanical properties representing the short-term behavior of rock mass (i.e., the elastoplastic parameters $G_M$, $C$, $\varphi$, $\psi$) were assumed known and constants.

As an example, the results of the prior and posterior distributions, as well as the posterior predictions issues from the BI, are presented in Figure 2. More precisely, the BI was conducted by using the best-fit parameters $\eta_M = 3.09$ (GPa.year) and $\beta = 0.32$ obtained from the deterministic calibration as the mean values of the prior distribution. By assuming the Gaussian function of the prior distribution, their chosen standard deviations were calculated from a chosen value of coefficient of variation COV = 30%. The obtained results show that the mean values of the posterior distribution of these two parameters were, respectively, $\eta_M = 3.08$(GPa.year) and $\beta = 0.35$, with the standard deviations 0.102(GPa.year) and 0.02.

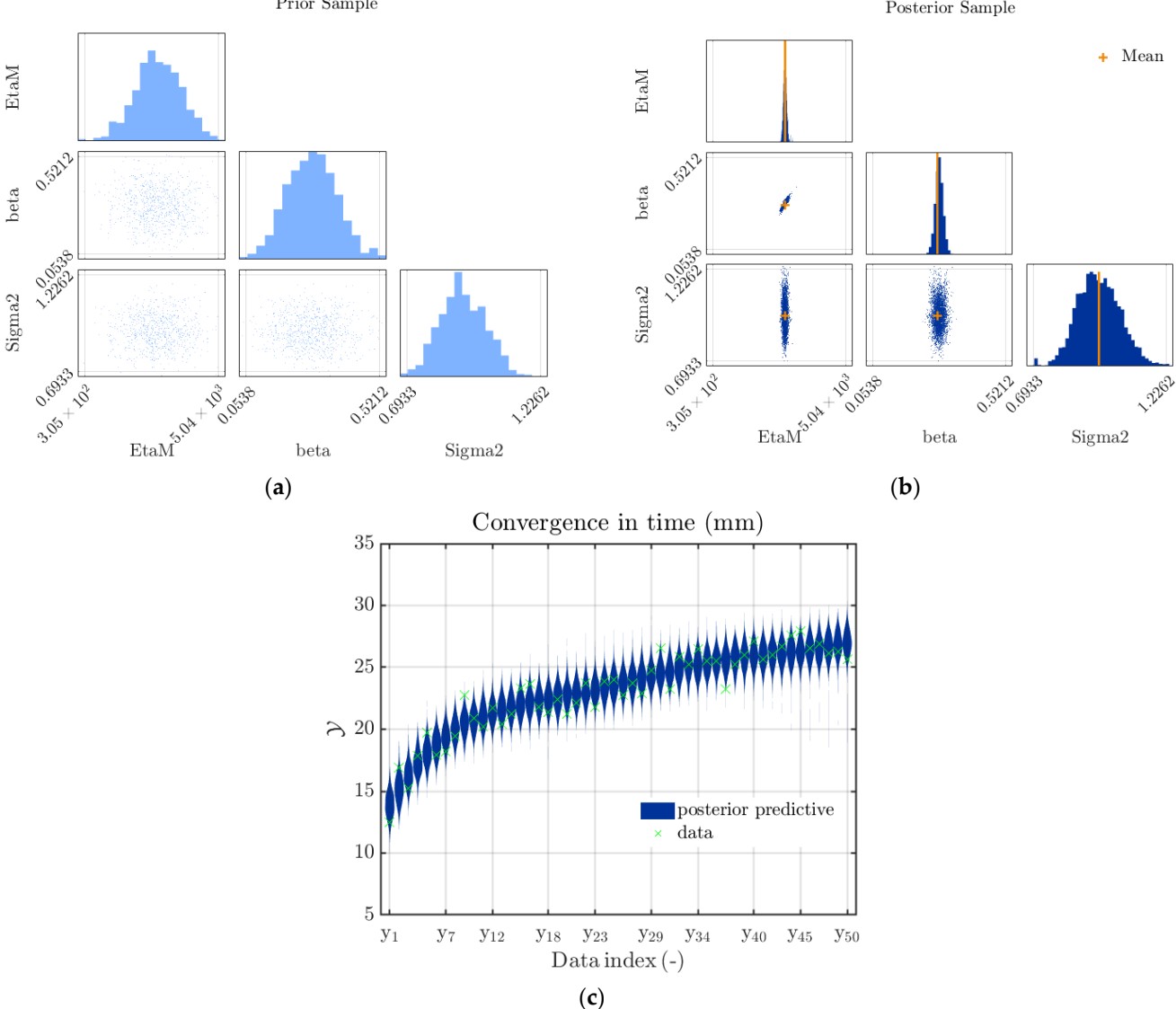

**Figure 2.** Results of the classical BI using the synthetic data of tunnel convergence in creep rock: (**a**) prior distribution; (**b**) posterior distribution of the time-dependent behavior parameters ($\eta_M$, $\beta$) of the FDVP model; (**c**) the posterior predictions.

The effect of prior distribution on the results of BI was then investigated. Different prior mean values of $\eta_M$ and $\beta$ were considered. This effect is highlighted in Figure 3,

in which the evolution of mean values of the posterior distribution of $\eta_M$ and $\beta$ were plotted as functions of their chosen prior mean values. As expected, the accuracy of the results provided by the classical BI would improve when the prior values approach the exact parameters. We can state from this investigation that the results of the deterministic inversion can provide an appropriate choice for the mean values of the prior distribution of the BI.

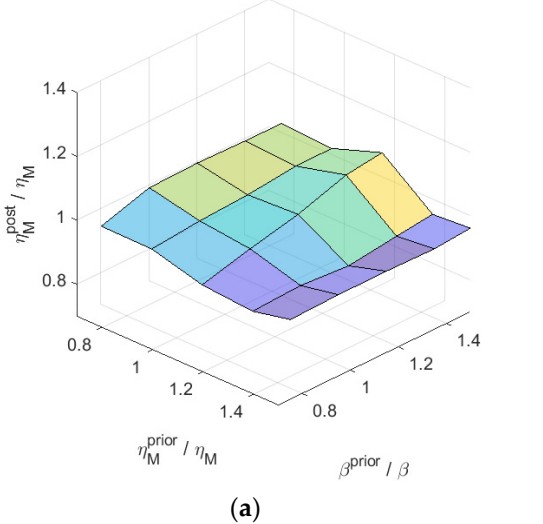
(**a**)

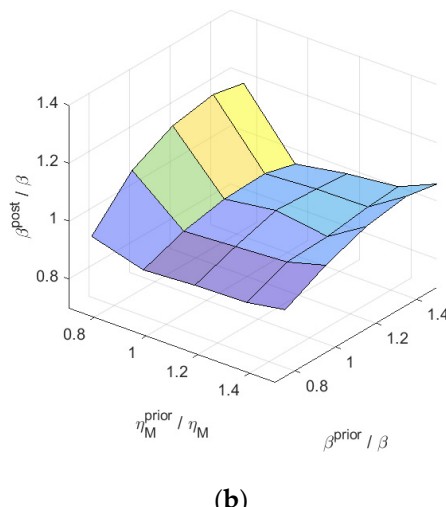
(**b**)

**Figure 3.** Mean values of the posterior distribution obtained from classical BI using different prior values: (**a**) parameter $\eta_M$; (**b**) parameter $\beta$ of the FDVP rock.

The influence of the additive noise magnitude was then found by changing its standard deviation values $\sigma_n$ on the results of the identified parameters. As illustrated in Figure 4, the mean values of the posterior distribution of the two parameters $\eta_M$ and $\beta$ matched well with their exact values when the additive noise was small. The higher magnitude of noise reduced the correctness of the identified parameters. More precisely, the difference between the posterior mean values and the exact parameters was more pronounced, and their corresponding standard deviations of the posterior functions were also higher when the additive noise magnitude was more important.

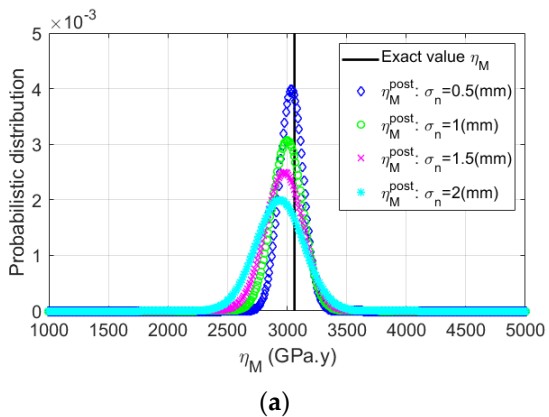
(**a**)

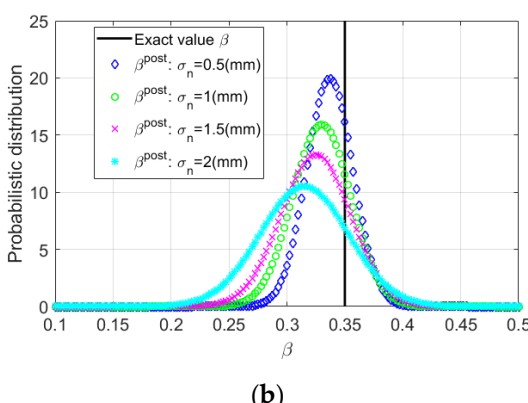
(**b**)

**Figure 4.** Posterior distribution of: (**a**) parameter $\eta_M$; (**b**) parameter $\beta$ of the FDVP rock as a function of additive noise magnitude $\sigma_n$.

The synthetic data generated by adding the random noise of Gaussian distribution with a mean of zero in the deterministic results of tunnel convergence presented, in effect, the ideal case of the homogeneous behavior of rock. This additive noise, known as the epistemic uncertainty, represents the lack of knowledge caused by measurement errors or a

small number of measurements. However, the rock formation is usually heterogeneous in nature, and the inherent randomness of rock properties contribute another source of uncertainty to the so-called aleatoric uncertainty.

In what it follows, both the additive noise (with $\sigma_n$ = 1 mm) and aleatoric uncertainty were taken in the synthetic data of tunnel convergence. For the generation of this latter uncertainty, the two parameters ($\eta_M$ and $\beta$) of the FDVP rock were assumed to be random, whose distributions are Gaussian, and the mean values are equal to the ones in Table 1 (i.e., $\mu_{\eta M}$ = 3.06 (GPa.year) and $\mu_\beta$ = 0.35). The variability of each parameter was characterized by a coefficient of variation (COV). For the sake of simplicity, the same COV = 15% was supposed for these two input parameters (i.e., their corresponding standard deviations were $\sigma_{\eta M}$ = 0.46(GPa.year) and $\sigma_\beta$ = 0.053). As an example, Figure 5 presents ten synthetic data sets of convergence determined at ten sections along the tunnel axis, using both aleatoric uncertainty and additive noise.

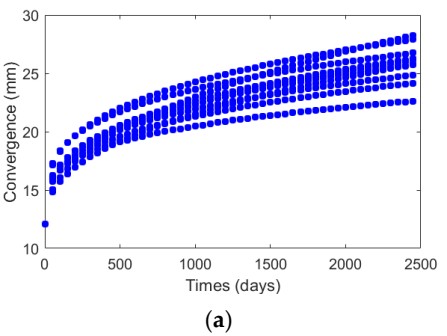 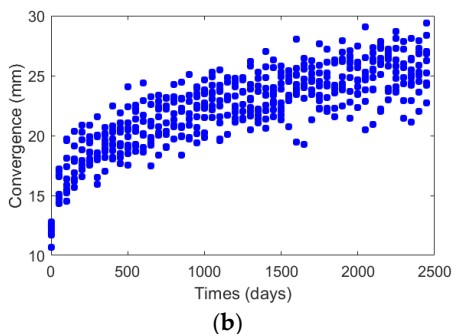

(**a**)        (**b**)

**Figure 5.** Synthetic data of tunnel convergence using: (**a**) aleatoric uncertainty; (**b**) both aleatoric and epistemic uncertainty.

The hierarchical BI technique, as described in the previous section, can be used now to solve this stochastic inversion problem. Following that, for each data set, the deterministic inversion was carried out to calibrate the two input parameters that were then chosen as the mean values of the prior distribution. The postprocessing by drawing samples from the posterior distribution obtained from the results of the classical BI conducted on each data set was undertaken to simulate the likelihood function, as defined in Equation (10).

Figure 6 illustrates the results of the hierarchical BI using different numbers of synthetic data sets ($N_s$ = 5, 10, 20, 50). Following that, the posterior distributions of the hyperparameters (i.e., the mean and standard deviation of the two parameters $\eta_M$ and $\beta$) presented a quite similar tendency. When increasing the number of data sets, the point estimates by hierarchical BI approached the true values of the mean and standard deviation of each parameter of the FDVP rock. Their correspondingly-estimated uncertainty also decreased, as expected, using the higher number sets of convergence.

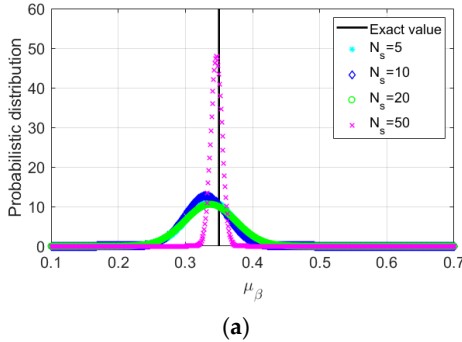 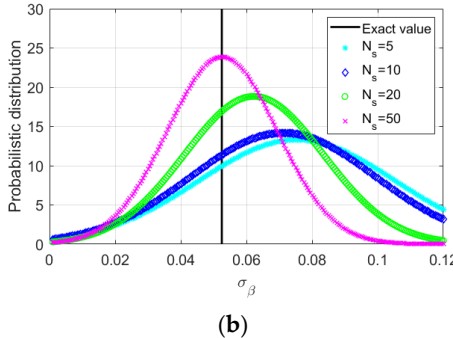

(**a**)        (**b**)

**Figure 6.** Posterior distribution of: (**a**) mean value; (**b**) standard deviation value of the two parameters $\eta_M$ and $\beta$ of FDVP rock determined by hierarchical BI using different numbers of synthetic data sets.

The results provided by the hierarchical BI were then compared with the ones calculated from the classical BI using different numbers of data sets. Note that, in this latter inversion process, the aleatoric uncertainty was no longer accounted for, but instead it was considered as another source that increases the epistemic uncertainty. Figure 7 captures the results of both methods. While the mean values of the two parameters $\eta_M$ and $\beta$ of FDVP rock can be determined quite well by the classical BI, their corresponding standard deviations were very far from the exact values and from the mean of posterior distribution evaluated by hierarchical BI. Consequently, as summarized in Table 2, the distribution of the two parameters $\eta_M$ and $\beta$ ranging from the minimum to the maximum values (which correspond to the lower and upper quantiles of 2.5% and 97.5%) were very different with respect to the exact results and the ones of the hierarchical BI.

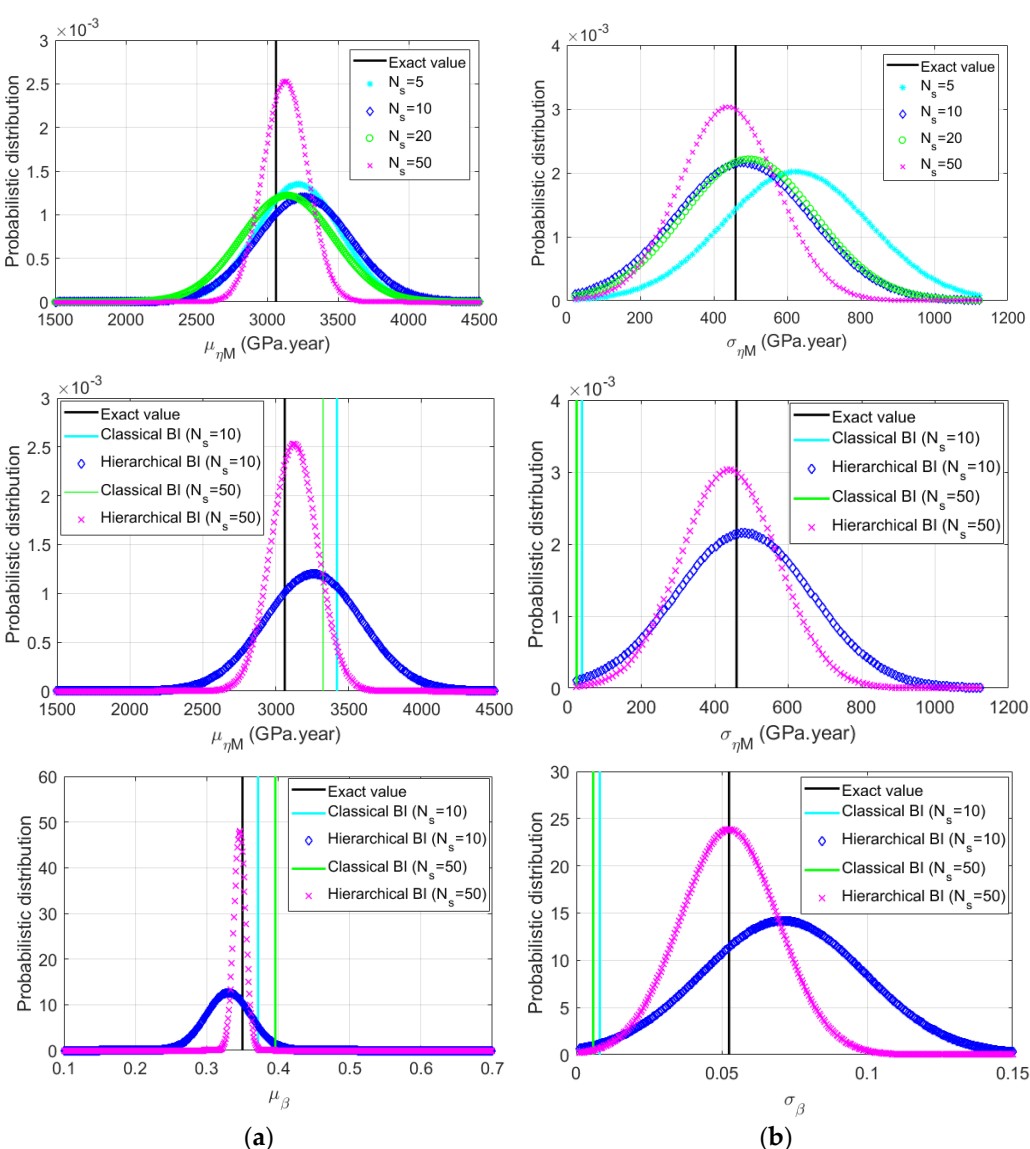

**Figure 7.** Posterior distribution of: (**a**) mean value; (**b**) standard deviation value of the two parameters $\eta_M$ and $\beta$ of FDVP rock using the classical and hierarchical BI.

**Table 2.** Minimum and maximum values (corresponding to lower quantile 2.5% and upper quantile 97.5%) of FDVP parameters using classical and hierarchical BI.

| Method | $\eta_M$ (GPa.year) | | $\beta$ | |
|---|---|---|---|---|
| | **Min** | **Max** | **Min** | **Max** |
| Exact | 2.16 | 3.96 | 0.25 | 0.45 |
| Classical BI ($N_s = 10$) | 3.34 | 3.50 | 0.35 | 0.39 |
| Classical BI ($N_s = 50$) | 3.28 | 3.37 | 0.38 | 0.41 |
| Hierarchical BI ($N_s = 10$) | 0.95 | 5.56 | 0.25 | 0.46 |
| Hierarchical BI ($N_s = 50$) | 1.45 | 4.80 | 0.16 | 0.53 |

## 4. Uncertainty of Time-Dependent Behavior of COx Claystone

The capability of the BI to quantify the uncertainty of the time-dependent behavior of creep rock was discussed in the previous part using synthetic data. More specifically, the efficiency of the BI was demonstrated when both the epistemic and aleatoric uncertainties could be accounted for. The BI was then applied in this section to quantify the uncertainty of viscoplastic properties of COx claystone using the real data of convergence of drifts in URL.

### 4.1. Description of the Numerical Model

It was observed that the excavation of drift in the COx claystone induced a fractured zone [2,3]. Specifically, for drifts parallel to the major horizontal stress $\sigma_H$, a dissymmetrical shape of the fractured zone was observed despite the quasi-isotropic stress state ($\sigma_v \approx \sigma_h \approx 12.5$ MPa) in the drift section [2,3]. The fracture zone was more developed in the horizontal direction, with an extension to about 1 time the diameter of the drift (Figure 8a). The convergence measurements also highlighted a higher convergence about two times in the horizontal direction, but the evolution in time of both the horizontal and vertical convergence seemed similar (Figure 8b) [2].

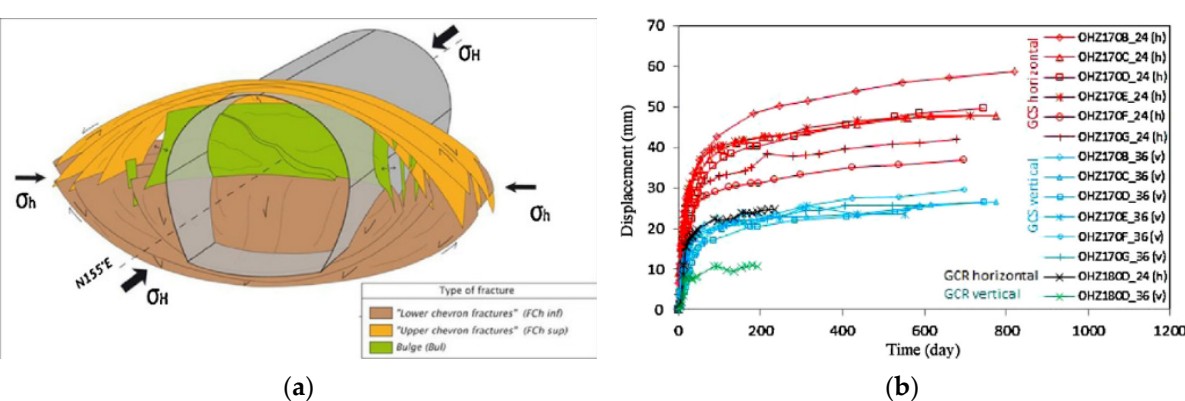

(**a**)          (**b**)

**Figure 8.** (**a**) Induced fracture network; (**b**) horizontal and vertical convergences observed in the drifts excavated following the direction of major horizontal stress [1].

To reproduce the observed phenomenon of the anisotropic convergence of drift, in this work, we adopted the simplified approach proposed in [10,11]. Following that, in the 2D plane strain model of the drift of 5.2 m of diameter, the dissymmetrical shape of the fractured zone was imposed (Figure 9a). More precisely, this fractured zone had an elliptical form extended around the drift to about 1 and 0.125 times the diameter, following the correspondingly horizontal and vertical direction. The symmetric conditions allowed us to model only one quarter of the section of drift. The normal stress that equals to the isotropic initial stress of 12.5MPa was imposed on the top and right boundaries found at 26 m with respect to the center of drift (i.e., about ten times the radius of the drift). At the

other boundaries (i.e., on the left and bottom boundaries) the displacement was fixed in the normal direction.

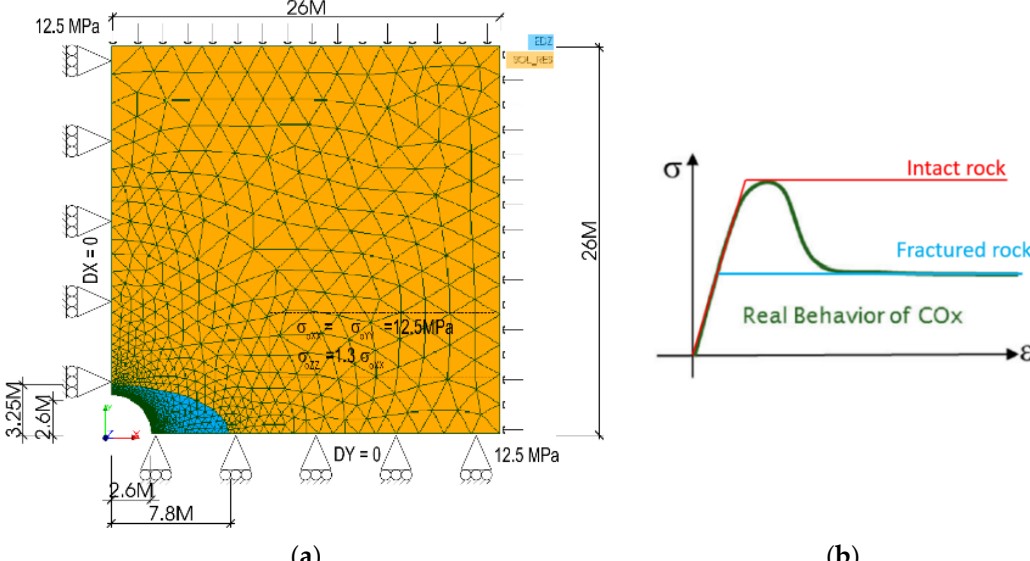

(**a**)                                          (**b**)

**Figure 9.** (**a**) Geometrical model with the elliptical fractured zone around the drift; (**b**) elastic-perfectly plastic Mohr Coulomb model of intact and fractured rocks.

The Mohr-Coulomb criterion was adopted to describe the instantaneous behavior of COx claystone. While the elastic properties were similar in the whole medium, the plastic properties of the intact rock and fractured zone that correspond to the peak and residual plastic parameters of the uniaxial compression stress-strain curve were different (Figure 9b). It was also assumed that both fractured and intact rock had the same time-dependent behavior that was characterized by the well-known viscoplastic Lemaitre model (see [12] for more details of this model). Thus, following the adopted approach, the coupled elastoplastic-viscoplastic model (characterized by the corresponding Mohr-Coulomb criterion and Lemaitre model) was considered in this study, while the predefined dissymmetrical shape of the fractured zone was the principal source to describe the anisotropy of drift's convergence. Totally, ten parameters were used in this model, which consist of: two elastic parameters (Young's modulus $E$ and Poisson's ratio $\nu$), three parameters of the plastic Mohr-Coulomb criterion of intact rock ($C_i$, $\varphi_i$, $\psi_i$) and of the fractured zone ($C_f$, $\varphi_f$, $\psi_f$), and three parameters of the viscoplastic Lemaitre model ($K$, $n$, $m$).

As an illustration purpose, Figure 10 highlights the evolution in time of horizontal and vertical convergences calculated correspondingly at the springline and crown of drift. These results were provided from the deterministic simulation, using the finite element *Code_Aster* with input parameters summarized in Table 3. While the elastoplastic parameters taken were similar to those in [11], the viscoplastic parameters of the Lemaitre model were taken from the deterministic calibration of convergence data captured at a section of the drift. The anisotropic convergence was well observed by the numerical simulation, which confirmed the capability of the simplified method by considering, explicitly, the dissymmetric shape of the fractured zone.

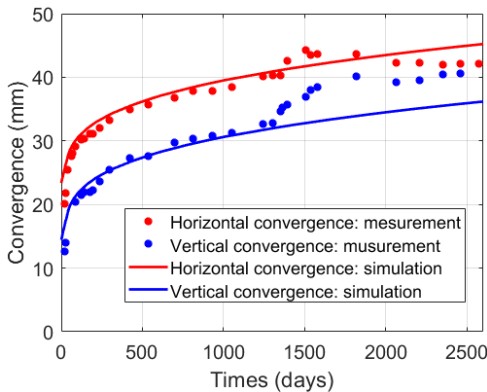

**Figure 10.** Evolution in time of horizontal and vertical convergences at a section of the drift: comparison of experimental and numerical simulation of the deterministic problem.

**Table 3.** Elasto-plastic properties of COx claystone.

| $E$ (GPa) | $v$ | $C_i$ (MPa) | $\varphi_i$ (°) | $\psi_i$ (°) | $C_f$ (MPa) | $\varphi_f$ (°) | $\psi_f$ (°) |
|---|---|---|---|---|---|---|---|
| 6.6 | 0.3 | 6 | 20 | 0 | 1 | 25 | 5 |

### 4.2. Results of the Bayesian Inversion and Discussions

To simplify the inversion procedure, herein, we considered the deterministic elastoplastic parameters that characterize the instantaneous behavior of the intact and fractured rocks to be known. Thus, the BI was conducted to quantify only the uncertainty of three random variables of the viscoplastic Lemaitre model, which described the time-dependent behavior of COx claystone. For this purpose, the horizontal and vertical convergences measured at six sections of drift, as shown in Figure 8b, were used throughout the BI process.

Both the classical and hierarchical BI were considered in the stochastic inversion to determine the mean and standard deviation of the viscoplastic Lemaitre parameters. More precisely, in the former BI technique, by ignoring the heterogeneous characteristic (i.e., the aleatoric uncertainty) of the COx viscoplastic properties, whole convergence data measured at six sections were used. Regarding the hierarchical BI, the classical BI was carried out to determine the mean and standard deviation of the viscoplastic Lemaitre parameter of each section of drift (Figure 11). Note that, in the classical BI, the results of the deterministic calibration process were chosen as the mean values of the prior distribution of these random parameters. Then, based on the postprocessing of the results obtained from these classical BI, we quantified, in the hierarchical BI process, the uncertainty of the hyperparameters (i.e., mean and standard deviation) of the Lemaitre parameters.

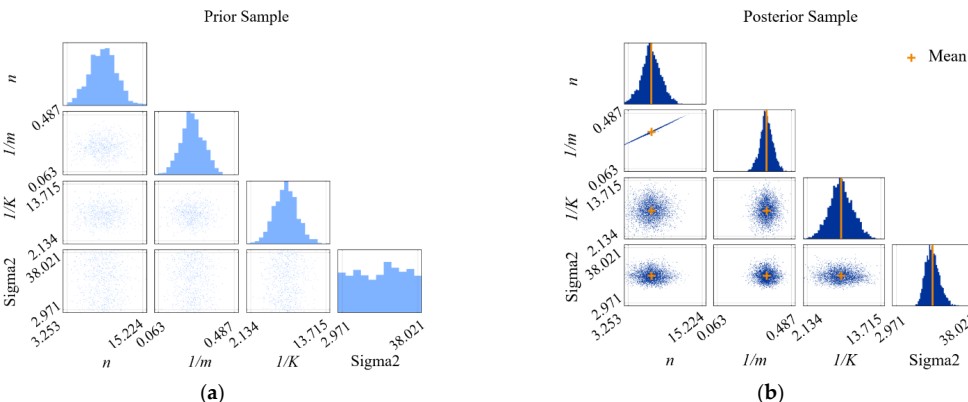

**Figure 11.** (**a**) Prior distribution; (**b**) posterior samples of the Lemaitre parameters (*n*, *1/m*, *1/K*) of the COx claystone determined by classical BI using convergence data of one section of drift.

Table 4 summarizes the results of the classical BI conducted using data of each section and of six sections of drift at the same time. Figure 12 presents the results of the mean and standard deviation of the viscoplastic Lemaitre parameters of the COx provided by the classical and hierarchical BI. The comparison of the two BI process showed an important difference of the obtained standard deviation of each viscoplastic parameter. The deterministic results calculated by the classical BI method were much higher than the mean value of the posterior distribution provided by the hierarchical BI. Regarding the corresponding mean value of each Lemaitre parameters, we stated that the difference was moderate.

**Table 4.** Mean and standard deviation (Std) of viscoplastic properties of COx claystone at different sections using the classical BI.

| Section | *n* | | *1/m* | | *1/K* (GPa$^{-1}$) | |
|---|---|---|---|---|---|---|
| | Mean | Std | Mean | Std | Mean | Std |
| OHZ170B | 9.22 | 1.78 | 0.26 | 0.040 | 8.25 | 1.95 |
| OHZ170C | 8.97 | 1.72 | 0.25 | 0.039 | 7.19 | 1.79 |
| OHZ170D | 8.94 | 1.69 | 0.24 | 0.038 | 7.15 | 1.78 |
| OHZ170E | 8.92 | 1.70 | 0.25 | 0.038 | 7.35 | 1.83 |
| OHZ170F | 8.48 | 1.55 | 0.23 | 0.035 | 7.07 | 1.78 |
| OHZ170G | 8.95 | 1.63 | 0.25 | 0.037 | 7.49 | 1.88 |
| Six sections | 8.66 | 1.82 | 0.23 | 0.041 | 8.36 | 2.16 |

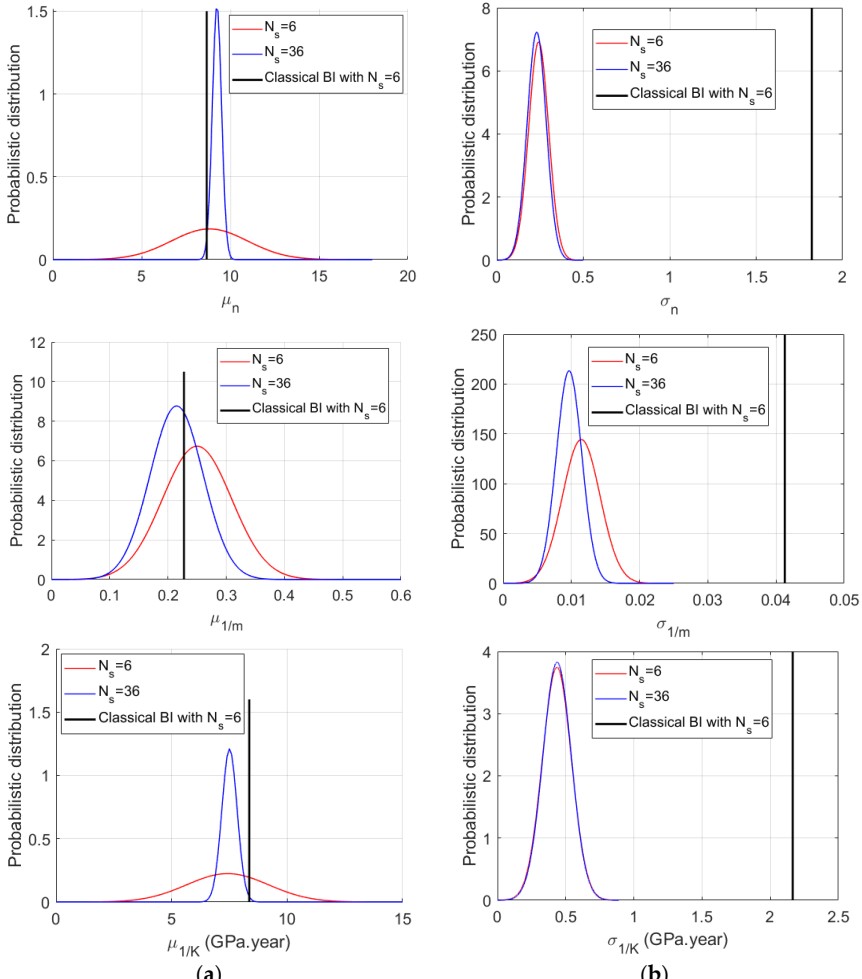

**Figure 12.** Posterior distribution of: (**a**) mean value; (**b**) standard deviation value of Lemaitre parameters of COx claystone determined by classical and hierarchical BI using the convergence data of six sections of drift.

To improve the results of the hierarchical BI, we proposed to increase the number of data sets for this inversion process. To this end, the horizontal convergence curve of one section can be gathered with the vertical convergence curve of another section. Thus, thirty-six convergence data sets ($N_s = 36$) can be generated for the hierarchical BI instead of six initial data sets ($N_s = 6$). As observed in Figure 12, the increase of the number of data sets reduced the uncertainty of the hyperparameters of the viscoplastic COx claystone, notably their mean values represented by a narrower posterior distribution. Finally, in Table 5, we summarized the ranging values evaluated at the 2.5% and 97.5% quantiles of the viscoplastic properties of host rock that were calculated from the two BI methods. The range of each parameter was reduced as expected by increasing the number of data sets.

**Table 5.** Minimum and maximum values (corresponding to the lower quantile 2.5% and the upper quantile 97.5%) of viscoplastic Lemaitre parameters using classical and hierarchical BI.

| Method | $n$ | | $1/m$ | | $1/K$ (GPa$^{-1}$) | |
|---|---|---|---|---|---|---|
| | Min | Max | Min | Max | Min | Max |
| Classical BI | 5.09 | 12.24 | 0.15 | 0.31 | 4.12 | 12.60 |
| Hierarchical BI ($N_s = 6$) | 3.91 | 13.74 | 0.10 | 0.40 | 2.68 | 12.17 |
| Hierarchical BI ($N_s = 36$) | 8.07 | 10.42 | 0.18 | 0.26 | 5.63 | 9.39 |

For the sake of clarity, the uncertainty quantification of the time-dependent behavior of COx claystone conducted in this work was based on different hypotheses. Firstly, the observed anisotropy of drift convergence was only reproduced in the simplified manner by imposing ad hoc an elliptical fracture zone around the drift. Secondly, the constant elastoplastic parameters of COx claystone were adopted in this study. These parameters played an important role in the distribution of the stress state in the host rock around the drift, and they could potentially affect the result of the probabilistic inversion of the Lemaitre viscoplastic properties. One can expect an improvement of this BI inversion by considering the short-term mechanical properties (e.g., the Young's modulus and Mohr Coulomb parameters of both intact and fractured rocks), being also random variables whose uncertainties must be quantified. In the other future work, the uncertainty of the time-dependent behavior of the host rock can be reconsidered, in which we use a more rigorous elasto-viscoplastic model of COx claystone that can reproduce accurately the anisotropic phenomenon of in-situ convergence measurements.

It is worth to note that, as an important contribution of this work, the effectiveness and the applicability of the BI to quantify both the epistemic and aleatoric uncertainty of creep rock were investigated. While most studies in the literature focus on the time-independent behavior of underground structures or are limited to time-dependent behavior by using the synthetic data of laboratory creep tests, the present study accounted for the in-situ convergence measurements in the inversion procedure. The structure effect made this considered problem more complex, so that the combination of the BI with the numerical simulation and metamodeling technique became a necessity. The efficiency of this combination was well demonstrated in this study, in which both the classical and hierarchical BI were considered. This process provided an interesting and useful tool that can be applied in different sites with different geological conditions.

## 5. Conclusions

In this work, the uncertainty quantification of the time-dependent behavior of COx claystone by BI was conducted using the real data of drift convergence. The process was firstly undertaken using the synthetic data that were generated from the analytical solution of the circular tunnel constructed in the fractional derivative viscoplastic rock. Gaussian additive noises with zero mean were artificially added in these synthetic data to represent the epistemic uncertainty related to the measurement error. The effects of the chosen prior distribution and magnitude of additive noise on the results provided by the classical BI

were highlighted. Particularly, the efficiency of the hierarchical BI was also demonstrated to solve the inverse problem, in which the aleatoric uncertainty was involved. The BI was then applied to quantify the uncertainty of the viscoplastic properties of COx claystone. To this end, real data of in-situ convergence measurements of the deep drift excavated in the major horizontal stress were considered. To simulate the complex time-dependent and anisotropic convergence of drift as observed in situ, we adopted the simplified approach to solve the deterministic problem. Following that, the combined elastoplastic-viscoplastic model using the Mohr Coulomb and Lemaitre models were chosen to characterize the instantaneous and time-dependent behavior of the host rock. Then, in the geometrical model, the same dissymmetric fractured zone, induced by excavation, was imposed. While the intact and fractured rocks had the same elastic and viscoplastic properties, their plastic properties were different. The uncertainty of the viscoplastic parameters of COx claystone quantified by hierarchical BI were provided, and the results can be improved when the number of data sets increases.

**Author Contributions:** Conceptualization, D.-P.D., D.H. and M.-N.V.; methodology, D.-P.D., N.-T.T. and D.H.; software, D.-P.D. and N.-T.T.; validation, D.-P.D., D.H., M.-N.V. and G.A.; formal analysis, D.-P.D. and D.H.; investigation, D.-P.D., N.-T.T. and D.H.; resources, D.-P.D., M.-N.V. and G.A.; data curation, M.-N.V. and G.A.; writing—original draft preparation, D.-P.D., N.-T.T. and D.H.; writing—review and editing, D.-P.D., D.H. and M.-N.V.; visualization, D.-P.D. and N.-T.T.; supervision, D.H., M.-N.V. and G.A.; project administration, D.-P.D. All authors have read and agreed to the published version of the manuscript.

**Funding:** This research received no external funding.

**Conflicts of Interest:** The authors declare no conflict of interest.

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
