# Peer review of "Time-Dependent Behavior of Callovo-Oxfordian Claystone for Nuclear Waste Disposal: Uncertainty Quantification from In-Situ Convergence Measurements"

_sustainability, doi:10.3390/su14148465_

Round 1

Reviewer 1 Report

This paper using the well-known Bayesian inference technique to study the uncertainty quantification of the creep behavior of COx claystone. Results are presented and discussed in consistent way. The results were in line with experimental observations and support their interpretation. I think this work is interesting and should be published with minor revisions. Nevertheless, there are some important aspects that need to be clarified before manuscript acceptance:

(1) The important understanding of this paper should be summarized in the abstract, rather than just introduce the significance of the research.

(2) The novelty of this work should be improved in the Introduction Section.

(3) Many of the pictures are not clear enough, the font needs to be bigger, such as Fig. 1, Fig. 2, Fig. 3, Fig. 4, Fig. 8, Fig. 9, Fig. 10, Fig. 11.

(4) In Section 4. Uncertainty of time-dependent behavior of COx claystone. I strongly recommend a more detailed modeling process should be supplemented, including model geometry, initial and boundary conditions, materials parameters (e.g., claystone, casing, gap, etc.).

(5) The applicability and effectiveness of the model in different site conditions need to be further discussed, thus the discussion section can be added in the revised manuscript.

Author Response

We thank the reviewer for his important remarks and recommendations.

This paper using the well-known Bayesian inference technique to study the uncertainty quantification of the creep behavior of COx claystone. Results are presented and discussed in consistent way. The results were in line with experimental observations and support their interpretation. I think this work is interesting and should be published with minor revisions. Nevertheless, there are some important aspects that need to be clarified before manuscript acceptance:

  • The important understanding of this paper should be summarized in the abstract, rather than just introduce the significance of the research.

Reply :

The important understanding was summerized in the abstract

  • The novelty of this work should be improved in the Introduction Section.

Reply :

We highlight the novelty of this work in the introduction section

  • Many of the pictures are not clear enough, the font needs to be bigger, such as Fig. 1, Fig. 2, Fig. 3, Fig. 4, Fig. 8, Fig. 9, Fig. 10, Fig. 11.

Reply :

The bigger font was used now in this revised version

  • In Section 4. Uncertainty of time-dependent behavior of COx claystone.I strongly recommend a more detailed modeling process should be supplemented, including model geometry, initial and boundary conditions, materials parameters (e.g., claystone, casing, gap, etc.).
  • The applicability and effectiveness of the model in different site conditions need to be further discussed, thus the discussion section can be added in the revised manuscript.

Reply :

These recommandation are quite similar as the ones of the second reviewer. Thus, in the section 4 : we add two sub-sections : the first one is dedicated to the description of the numerical model whislt in the second one, we present the results and discussions.

Reviewer 2 Report

The research of geotechnical issues in a clay-rock considered for HLW disposal is a relevant topic. The objective of the paper is clear: to quantify the uncertainty of open tunnel convergence after excavation due to stress redistribution. Besides short term processes time dependent behaviour of Callovo-Oxfordian claystone plays a role too. Understanding and uncertainty quantification are necessary for justification of safety of the repository. The methodology presented is followed by its demonstration on synthetic and real data. The conclusions are made about the uncertainty quantification using classical and hierarchical Bayesian inference. Results are presented and discussed in consistent way.

Nevertheless, there are some important aspects that need to be clarified before manuscript acceptance:

1. From the current description it is not clear if the uncertainty quantification with proposed methodology requires a certain number of visco elasto-plastic model runs or it is based on some constructed samples or numerical results of metamodel emulating visco elasto-plastic model output. If visco-elasto-plastic model was used does it require the input parameters to be sampled in particular way?

 2. I would suggest to introduce the behaviour, convergence of COx, description about fractured zone around the tunnel, its size, anisotropy, different numerical models proposed for description of processes right after “Introduction”. Then Chapter 4 should contain 1) the description of the numerical model used in this study: geometry, material data, initial and boundary conditions, simulation time, computer code(s) used; 2) then results and discussion.

Some other comments and suggestions are listed below:

1.     Line 28: …, the IPCC report on global …

Please provide full name before abbreviation.

2.      Line 30: Besides the technical and human challenge that this production implies, the nuclear waste management presents in itself a challenge because the long term behaviour of nuclear waste disposals is yet a very dynamic field of studies.

I would not agree that nuclear energy production is a technical and human challenge. Nuclear power plants are being safely operated over decades. There is a worldwide practise of low and intermediate radioactive waste management. Yes, there is no operating geological disposal facility for spent nuclear fuel in the world yet, but it is under construction in Finland, the license application for its operation has been submitted to regulatory.

 Please consider revision.

3.      Line 152: In addition, to reduce the computational cost of the numerical simulation, the well-known Kriging metamodeling technique (see [6, 7, 10, 11] for details) is also used to approximate the structure response.

What response was modelled with “Kriging metamodeling technique” in this study? Please clarify.

4.      Equations 11-16. The units for the parameters should be indicated.

5.      Equation 16 contain angle φ. Does it correspond to friction angle indicated as φ in line 180?

6.      Line 192: Totally, a data set with 50 measurements of tunnel convergence are artificially generated in the range of 2500 days.

If tunnel convergence data are artificially generated via analytical solution, term “measurements” sounds misleading.

7.      Figure 1.

Consider adding term “calculated” for caption b) convergence on the surface of tunnel without and with additive noise: calculated convergence on the surface of tunnel without and 195 with additive noise.

8.      Line 208: (GPa.year).

Please clarify the units: GPa∙year?

9.      Line 383: Thus, BI is conducted to quantify only the uncertainty of three random variables of the viscoplastic Lemaitre model which describe the time-dependent behavior of COx claystone.

Are these variables sampled randomly? From what PDF?

Author Response

We would like to thank the reviewer for his useful remarks and recommendations.

The research of geotechnical issues in a clay-rock considered for HLW disposal is a relevant topic. The objective of the paper is clear: to quantify the uncertainty of open tunnel convergence after excavation due to stress redistribution. Besides short term processes time dependent behaviour of Callovo-Oxfordian claystone plays a role too. Understanding and uncertainty quantification are necessary for justification of safety of the repository. The methodology presented is followed by its demonstration on synthetic and real data. The conclusions are made about the uncertainty quantification using classical and hierarchical Bayesian inference. Results are presented and discussed in consistent way.

Nevertheless, there are some important aspects that need to be clarified before manuscript acceptance:

  1. From the current description it is not clear if the uncertainty quantification with proposed methodology requires a certain number of visco elasto-plastic model runs or it is based on some constructed samples or numerical results of metamodel emulating visco elasto-plastic model output. If visco-elasto-plastic model was used does it require the input parameters to be sampled in particular way?

Reply:

The authors thank the reviewer for this important question. In the BI process, the Kriging metamodel is constructed to approximate the tunnel convergence. For this aim, 240 random samples were generated by the Latin Hypercube Sampling technique. For each sample, the structure response (i.e., the tunnel convergence) was evaluated (by using the analytical solutions in the case of tunnel in FDVP rock or numerical simulation of drift in COx claystone).

This explanation was added in this revised version for the sake of clarity.

  1. I would suggest to introduce the behaviour, convergence of COx, description about fractured zone around the tunnel, its size, anisotropy, different numerical models proposed for description of processes right after “Introduction”. Then Chapter 4 should contain 1) the description of the numerical model used in this study: geometry, material data, initial and boundary conditions, simulation time, computer code(s) used; 2) then results and discussion.

Reply:

This recommendation was taken into account in this revised version. Following that, the description of the fractured zone and different numerical models were placed in the introduction section.  And in the section 4 : we add two sub-sections : the first one is dedicated to the description of the numerical model whislt in the second one, we present the results and discussions.

Some other comments and suggestions are listed below:

  1.    Line 28: …, the IPCC report on global …

Please provide full name before abbreviation.

Reply:

The full name “The Intergovernmental Panel on Climate Change of IPCC was provided in this version.

  1. Line 30: Besides the technical and human challenge that this production implies, the nuclear waste management presents in itself a challenge because the long term behaviour of nuclear waste disposals is yet a very dynamic field of studies.

I would not agree that nuclear energy production is a technical and human challenge. Nuclear power plants are being safely operated over decades. There is a worldwide practise of low and intermediate radioactive waste management. Yes, there is no operating geological disposal facility for spent nuclear fuel in the world yet, but it is under construction in Finland, the license application for its operation has been submitted to regulatory.

 Please consider revision.

Reply:

This remark was considered in this version. Following that the sentence was changed as “Whilst practise of low and intermediate radioactive waste management has been largely conducted, the long-term behavior of nuclear waste disposals is yet a very dynamic field of studies »

  1. Line 152: In addition, to reduce the computational cost of the numerical simulation, the well-known Kriging metamodeling technique (see [6, 7, 10, 11] for details) is also used to approximate the structure response.

What response was modelled with “Kriging metamodeling technique” in this study? Please clarify.

Reply:

The Kriging metamodel is constructed to approximate the tunnel convergence.

  1. Equations 11-16. The units for the parameters should be indicated.

Reply:

      The units of all necessary parameters were precised in tables 1 and 3. Only the convergence and radius in Equation (11) have the units in meter (m) whilst all the other parameters in Eqs (12 to 16) are dimensionless.

  1. Equation 16 contain angle φ. Does it correspond to friction angle indicated as φ in line 180?

      Reply:

      Yes, these parameters are the same that present the friction angle of rock mass. In this revised version, the angle φ is used in the whole paper.

  1. Line 192: Totally, a data set with 50 measurements of tunnel convergence are artificially generated in the range of 2500 days.

If tunnel convergence data are artificially generated via analytical solution, term “measurements” sounds misleading.

Reply:

The term “measurements” was deleted in this version

  1. Figure 1.

Consider adding term “calculated” for caption b) convergence on the surface of tunnel without and with additive noise: calculated convergence on the surface of tunnel without and with additive noise.

Reply:

The word “calculated” was added for capture b) in Figure 1 of this revised version

  1. Line 208: (GPa.year).

Please clarify the units: GPa∙year?

Reply:

This is the classical unit of the rheological parameter (i.e., the dashpot) of viscoelastic rock. One can use Pa.s or GPa.s or even GPa.year.

  1. Line 383: Thus, BI is conducted to quantify only the uncertainty of three random variables of the viscoplastic Lemaitre model which describe the time-dependent behavior of COx claystone.

Are these variables sampled randomly? From what PDF?

Reply:

Yest these variables are randomly sampled. In the BI both the prior and posterior distribution functions are supposed normal (or Gaussian) function.
